# MixPro: Data Augmentation with MaskMix and Progressive Attention Labeling for Vision Transformer

**Qihao Zhao[1], Yangyu Huang[2], Wei Hu[1]\*, Fan Zhang[1]†, Jun Liu[3]**
[1]Beijing University of Chemical Technology, China [2]Microsoft Research Asia, China
[3]Singapore University of Technology and Design, Singapore

## Abstract

The recently proposed data augmentation TransMix employs attention labels to help visual transformers (ViT) achieve better robustness and performance. However, TransMix is deficient in two aspects: 1) The image cropping method of TransMix may not be suitable for vision transformer. 2) At the early stage of training, the model produces unreliable attention maps. TransMix uses unreliable attention maps to compute mixed attention labels that can affect the model. To address the aforementioned issues, we propose MaskMix and Progressive Attention Labeling (PAL) in image and label space, respectively. In detail, from the perspective of image space, we design MaskMix, which mixes two images based on a patch-like grid mask. In particular, the size of each mask patch is adjustable and is a multiple of the image patch size, which ensures each image patch comes from only one image and contains more global contents. From the perspective of label space, we design PAL, which utilizes a progressive factor to dynamically re-weight the attention weights of the mixed attention label. Finally, we combine MaskMix and Progressive Attention Labeling as our new data augmentation method, named MixPro. The experimental results show that our method can improve various ViT-based models at scales on ImageNet classification (73.8% top-1 accuracy based on DeiT-T for 300 epochs). After being pre-trained with MixPro on ImageNet, the ViT-based models also demonstrate better transferability to semantic segmentation, object detection, and instance segmentation. Furthermore, compared to TransMix, MixPro also shows stronger robustness on several benchmarks.

## 1 Introduction

Transformers (Vaswani et al., 2017) have revolutionized the natural language processing (NLP) field and have recently inspired the emergence of transformer-style architectures in the computer vision (CV) field, such as Vision Transformer (ViT) (Dosovitskiy et al., 2020). These methods design with competitive results in numerous CV tasks like image classification (Touvron et al., 2021a; Yuan et al., 2021; Wang et al., 2021; Liu et al., 2021; Touvron et al., 2021b; Ali et al., 2021), object detection (Fang et al., 2021; Dai et al., 2021; Carion et al., 2020; Zhu et al., 2020) and image segmentation (Strudel et al., 2021; Wang et al., 2021; Liu et al., 2021). Previous research has discovered that ViT-based networks are difficult to optimize and can easily overfit to training data with many images (Russakovsky et al., 2015), resulting in a significant generalization gap in the test data. To improve the generalization and robustness of the model, the recent works (Dosovitskiy et al., 2020; Touvron et al., 2021a; Yuan et al., 2021; Wang et al., 2021; Liu et al., 2021; Touvron et al., 2021b; Ali et al., 2021) employ data augmentation (Zhang et al., 2017) and regularization techniques (Szegedy et al., 2016) during training. Among them, the mixup-base methods such as Mixup (Zhang et al., 2017), CutMix (Yun et al., 2019) and TransMix (Chen et al., 2021) are implemented to improve the generalization and robustness of ViT-based networks. For the CutMix, patches are cut and pasted among training images where the ground truth labels are also mixed proportionally to the area of the patches. Furthermore, TransMix based on CutMix considers that not all pixels are created equal.

---

\*Corresponding author. huwei@buct.edu.cn
†Fan Zhang is with the College of Information Science and Technology and the Interdisciplinary Research Center for Artificial Intelligence, Beijing University of Chemical Technology, Beijing 100029.

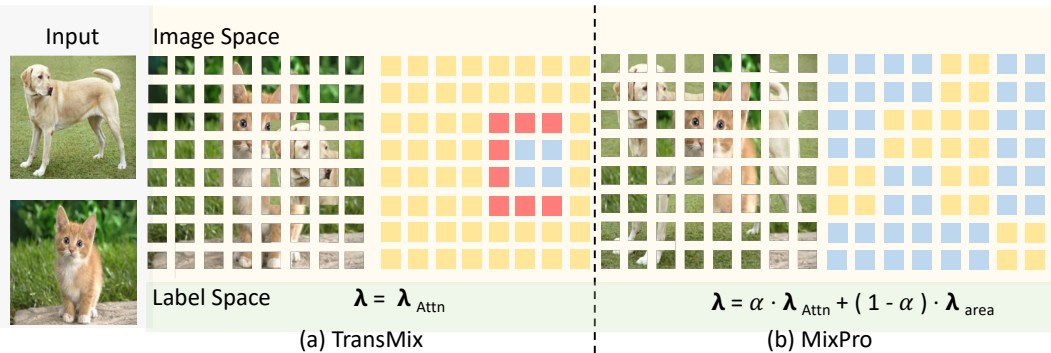

Figure 1: Comparison between TransMix (Chen et al., 2021) (a) and our proposed MixPro (b). 1) For image space, TransMix shares the same cropped region with CutMix (Yun et al., 2019), which results in patches containing different regions from the two images (patches colored red). Differently, as shown on the right of the figure, MixPro mixes patches using a patch-like mask. The size of the mask patches is the multiple of the image patches. This enables each patch of the mixed image to come from only one image (patches colored yellow and blue). 2) For label space, TransMix computes $\lambda$ by $\lambda_{attn}$. In contrast, we propose progressive attention labeling that dynamically re-weights $\lambda_{area}$ and $\lambda_{attn}$ using a progressive factor ($\alpha$).

Then it exploits ViT's attention map to re-assign the weight to the mixed label rather than applying the proportion of the cropped image area.

Nevertheless, TransMix (Chen et al., 2021) has the following drawbacks for vision transformer: (1) TransMix and CutMix share the same region-level cropped images as input. However, ViT-based models naturally have global receptive fields from self-attention (Dosovitskiy et al., 2020), so the region-based mixed images may provide insufficient image contents. (More details are in Sec. A.1.) (2) As shown in Fig. 1 (a)TransMix provides cropped patches with sharp rectangular borders that are clearly distinguishable from the background (viewed as red patches). The ViT-based models may naturally be curious about the cropped patch and then pay attention to that patch, resulting in a basic weight of attention regardless of whether the patch contains useful information (Chen et al., 2021). (3) TransMix employs attention map to re-weight the confidence for the mixed targets. However, attention maps may not always be **reliable** during the training process. For example, at the beginning of the training, the model has no representation capability, and the attention maps gained are unreliable. In addition, it is possible to obtain difficult samples using massive data augmentation strategies, and the attention map is also unreliable. At this point, reassigning the mixed labels utilizing a low-confidence attention map will generate noisy mixed labels.

To this end, we propose a novel data augmentation method, *MixPro*, to tackle the above issues from the perspective of image space and label space, respectively. Our approach is presented in Fig. 1 (b). In detail, from the perspective of image space, we designed MaskMix, which is inspired by the mask strategy of MAE (He et al., 2022). Our MaskMix replaces the masked patches of one image with visible patches of another image to create a mixed image. In particular, the scale of each mask patch is adjustable and is a multiple of the image patch size. In this way, each image patch comes from only one image (viewed as yellow and blue patches in the figure). In addition, the mask decomposition of pictures can take into account both region and global contents. From the perspective of label space, we designed Progressive Attention Labeling (PAL), which utilizes a progressive factor ($\alpha$) to dynamically re-weight the attention weight of the mixed attention label. The progressive factor ($\alpha$) provides an indirect measure of the confidence of the attention map for the mixed sample, to trade-off the attention proportional weight ($\lambda_{attn}$) and the area proportional weight ($\lambda_{area}$). In conclusion, we combine Mask**Mix** and **Pro**gressive Attention Labeling, namely MixPro, as our data augmentation strategy to improve the generalization and robustness of ViT-based models.

In experiments, we demonstrate extensive evaluations of MixPro on various ViT-based models and tasks. MixPro exhibits greater performance gains than TransMix for all listed ViT-based models. Notably, MixPro can further bring an improvement of 0.9% for DeiT-S. Moreover, we demonstrate that if the model is first pretrained with MixPro on ImageNet, the superiority can be further transferred onto downstream tasks including object detection, instance segmentation, and semantic segmentation.

In terms of robustness, compared to TransMix, MixPro also shows stronger robustness on three different benchmarks.

Overall, we summarize our contributions as follows:

- We propose a new data augmentation method, MixPro, to address the shortcomings of TransMix from the perspective of image space and label space, respectively.

- From the perspective of image space, MixPro ensures that each image patch comes from only one image and contains more global contents. From the perspective of label space, MixPro utilizes a progressive factor to dynamically re-weight the attention weight of the mixed attention label.

- In experiments, we demonstrate extensive evaluations of MixPro on various ViT-based models and downstream tasks. It boosts Deit-T achieving 73.8% and Deit-S achieving 81.3% on ImageNet-1K. Furthermore, compared to TransMix, MixPro also shows stronger robustness on three different benches.

## 2  RELATED WORK

**Vision Transformers (ViTs).** Transformers were initially proposed for sequence models such as machine translation (Vaswani et al., 2017). Inspired by the success of transformers in NLP tasks, Vision Transformer (ViT) (Dosovitskiy et al., 2020) attempted to apply transformers for image classification by treating an image as a sequence of patches with excellent results compared to even state-of-the-art convolutional networks. DeiT (Touvron et al., 2021a) further extended ViT using a novel distillation approach and a powerful training recipe. Based on the success of ViT, numerous ViT-based models have emerged (Yuan et al., 2021; Wang et al., 2021; Liu et al., 2021; Touvron et al., 2021b; Ali et al., 2021; Yang et al., 2021). PVT (Wang et al., 2021) developed a progressive shrinking pyramid and a spatial-reduction attention layer to obtain high-resolution and multi-scale feature maps under limited computation/memory resources. XCIT (Ali et al., 2021) built their models with cross-covariance attention as its core component and demonstrate the effectiveness and generality of our models on various computer vision tasks. Swin (Liu et al., 2021) proposed the shifted window-based self-attention and showed the model was effective and efficient on vision problems. These ViT-based models have achieved good results not only for classification tasks (Deng et al., 2009), but also for object detection (Fang et al., 2021; Dai et al., 2021; Carion et al., 2020; Zhu et al., 2020) and image segmentation (Strudel et al., 2021; Wang et al., 2021; Liu et al., 2021) tasks. However, these ViT-based models rely on MixUp-based data augmentation to enhance generalization in case of insufficient data.

**MixUp-based data augmentation.** Mixup (Zhang et al., 2017) was the first work to propose interpolation of two images and their labels to augment the training data. Subsequent variants of the Mixup appeared (Yun et al., 2019; Chen et al., 2021; Verma et al., 2019; Baek et al., 2021; Cascante-Bonilla et al., 2021; Kim et al., 2020; Walawalkar et al., 2020; Uddin et al., 2020). They can be mainly divided into two groups: global image mixture (e.g. Manifold Mixup (Verma et al., 2019), GridMix (Baek et al., 2021), TokenMix (Liu et al., 2022), and PatchMix (Cascante-Bonilla et al., 2021)), and local/region image mixture (e.g.CutMix (Zhang et al., 2017), Puzzle-Mix (Kim et al., 2020), Attentive-CutMix (Walawalkar et al., 2020), SaliencyMix (Uddin et al., 2020), and MixToken (Jiang et al., 2021)). Manifold Mixup (Verma et al., 2019) proposed to train neural networks on interpolations of hidden representations. GridMix (Baek et al., 2021) was composed of two local constraints: local data augmentation by grid-based image mixing and local patch mapping constrained by patch-level labels. In CutMix (Yun et al., 2019), patches are cut and pasted among training images where the ground truth labels are also mixed proportionally to the area of the patches. Attentive-CutMix (Walawalkar et al., 2020) enhanced CutMix by choosing the most descriptive regions based on the intermediate attention maps from a feature extractor, which enables searching for the most discriminative parts in an image. MixToken (Jiang et al., 2021) is a modified version of CutMix operating on the tokens after patch embedding. Among them, MixUp and CutMix are most successful in improving the performance of ViTs (Touvron et al., 2021a). Furthermore, TransMix (Chen et al., 2021) based on CutMix continues to improve the generalization and robustness of ViT-based models by reassigning the ground truth labels with transformer attention guidance. TokenMix (Liu et al., 2022) utilizes a pre-trained teacher model to generate attention maps for guiding the mixed labels.

All in all, our proposed MixPro differs significantly from the above MixUp-based methods. First of all, CutMix, MixToken and TransMix focus on region contents, whereas MixPro mixes images with adjustable mask patches and may contain more global contents. Secondly, TransMix mixes image patches may generate noisy attention maps, while MixPro generates clean attention maps for mixing labels. The last and most important one is that, from the viewpoint of label space, TransMix neglects the fact that the model attention map during training is not always reliable. MixPro solves this issue by using progressive factors to dynamically recalculate the attention weight of the mixed attention label without a pre-trained model to generate it.

## 3 METHOD

### 3.1 BACKGROUND

**Multi-head self-Attention.** Multi-head self-Attention as introduced by ViTs (Dosovitskiy et al., 2020), firstly divides and embeds an image $\mathbf{x} \in \mathbb{R}^{W \times H \times 3}$ to patch tokens $\mathbf{x}_{patches} \in \mathbb{R}^{N \times d}$, where $N$ is the number of tokens, each of dimensionality $d$. It aggregates the global information by a class token $\mathbf{x}_{cls} \in \mathbb{R}^{1 \times d}$. Then ViTs operate on the patch embedding $\mathbf{z} = [\mathbf{x}_{cls}, \mathbf{x}_{patches}] \in \mathbb{R}^{(1+N) \times d}$. Given a Transformer with $h$ attention heads and input patch embedding z, the class attention for each head can be formulated as:

$$\mathbf{q} = \mathbf{x}_{cls} \cdot \mathbf{W}_q, \tag{1}$$

$$\mathbf{k} = \mathbf{z} \cdot \mathbf{W}_k, \tag{2}$$

$$\mathbf{A}' = Softmax(\mathbf{q} \cdot \mathbf{k} / \sqrt{d/h}), \tag{3}$$

$$\mathbf{A} = \{A'_{0,i} | i \in [1, N]\}, \tag{4}$$

where $\mathbf{W}$ are weight matrices for q and k . $\mathbf{A} \in [0,1]^N$ is the attention map from the class token to the image patch tokens. It is summarized as which patches are most valuable for the final classifier. When there are multiple heads in the attention, we obtain them by simply averaging over all heads of the attention.

**TransMix.** TransMix (Chen et al., 2021) calculates $\lambda_{attn}$ (the proportion for mixing two labels) with the attention map $\mathbf{A}$ and a binary mask $\mathbf{M} \in \{0,1\}^{W \times H}$ from CutMix (Yun et al., 2019):

$$\lambda_{attn} = \mathbf{A} \cdot \downarrow (\mathbf{M}), \tag{5}$$

where $\downarrow (\cdot)$ denotes the nearest-neighbor interpolation downsampling that can transform the original $\mathbf{M}$ from $H \times W$ into $N$ pixels.

### 3.2 MIXPRO

Our approach, MixPro, consists of MaskMix and progressive attention labeling to improve the performance and robustness of ViT-based models from the viewpoint of image space and label space, respectively. Next, we will describe these two methods in detail.

**MaskMix.** Let $x \in \mathbb{R}^{W \times H \times C}$ denote a training image and let $y$ be its corresponding label. ViT (Dosovitskiy et al., 2020) regularly partitioned a high-resolution image $x$ into $N$ patches of a fixed size of $P_{image} \times P_{image}$ pixels ($N = W/P_{image} \times H/P_{image}$) and then embeds these to tokens. In our method, we first divide the grid mask $\mathbf{M} \in \{0,1\}^{W \times H}$ into $S = W/P_{mask} \times H/P_{mask}$ patches, resulting in each mask patch region of size $P_{mask} \times P_{mask}$. In particular, to make each image patch come from only one image, we ensure that the size of mask patches ($P_{mask}$) is a multiple of the image patch size ($P_{image}$). Then we select $\lfloor S \cdot \tau \rceil$ regions to mask, where

$$\tau = Beta(\beta, \beta), \tag{6}$$

and the value in the selected mask region is set to 1. In all our experiments, we follow CutMix (Yun et al., 2019) set $\beta$ to 1, the $\tau$ is sampled from the uniform beta distribution Beta(1,1). With the mask

$\mathbf{M}$, for samples $x_i$, $x_j$ and their corresponding labels $y_i$, $y_j$, we utilize the mask $\mathbf{M} \in \{0,1\}^{W \times H}$ to generate their mixing samples $(\widetilde{x}, \widetilde{y})$ :

$$\widetilde{x} = \mathbf{M} \odot x_i + (\mathbf{1} - \mathbf{M}) \odot x_j, \widetilde{y} = \lambda_{area} \odot y_i + (1 - \lambda_{area}) \odot y_j, \tag{7}$$

where $\lambda_{area} = \sum_{i=1}^{W} \sum_{i=1}^{H} \mathbf{M}(i,j)/(W \times H)$, binary mask filled with ones is represent as $\mathbf{1}$ and $\odot$ is element-wise multiplication. In this way, as illustrated in Fig.1 (b), image patches are aligned with mask patches when mixing images to ensure that each image patch comes from only one image.

**Progressive Attention Labeling.** However, the attention map $\mathbf{A}$ may not always be accurate for each mixed sample. At the early stage of the training, models have poor representational ability, and the attention map $\mathbf{A}$ may not be reliable for some difficult mixed samples. At this point, the mixed labels calculated with $\lambda_{attn}$ would limit the performance of ViT.

Since blindly using attention maps to reassign mixed labels is not reliable. The focus is on designing a progressive factor $(\alpha)$, which could provide an indirect measure of the confidence of the attention map for the mixed sample, to trade-off the attention proportional weight $(\lambda_{attn})$ and the area proportional weight $(\lambda_{area})$. For neural networks with ground-truth distributions, cross-entropy measures the epistemic uncertainty of the model (Kendall & Gal, 2017). The more certain the model is, the more reliable the attention map is obtained. There, we employ the cosine similarity of the model output to the ground-truth labels as a proxy of cross-entropy to measure whether a mixed sample can obtain a high-confidence attention map. We calculate the cosine similarity, i.e.,

$$\mathbf{d}(\mathbf{p}, \widetilde{\mathbf{y}}) = \frac{\mathbf{p} \cdot \widetilde{\mathbf{y}}^{\top}}{||\mathbf{p}|| \cdot ||\widetilde{\mathbf{y}}||}, \tag{8}$$

where $\mathbf{p}$ is the model's output softmax probability. Since both $\mathbf{p}$ and $\widetilde{\mathbf{y}}$ are non-negative vectors, the range of $\mathbf{d}$ is $\in [0,1]$. Ideally, the cosine similarity $\mathbf{d}$ closer to 1, the samples better the network fits the mixed sample and it can also produce a high-confidence attention map. Therefore, we use the cosine similarity as our progressive factor $\alpha$,

and the $\lambda$ written as: 
$$\lambda = \alpha \cdot \lambda_{attn} + (1 - \alpha) \cdot \lambda_{area}, \tag{9}$$

where $\lambda$ is the proportion to instead of $\lambda_{area}$ in Eq. (7) for mixing two labels. So far, we propose Progressive Attention Labeling (PAL), which employs the progressive factor $(\alpha)$ to dynamically re-weight the attention weight of the mixed attention label.

## 3.3 DISCUSSION

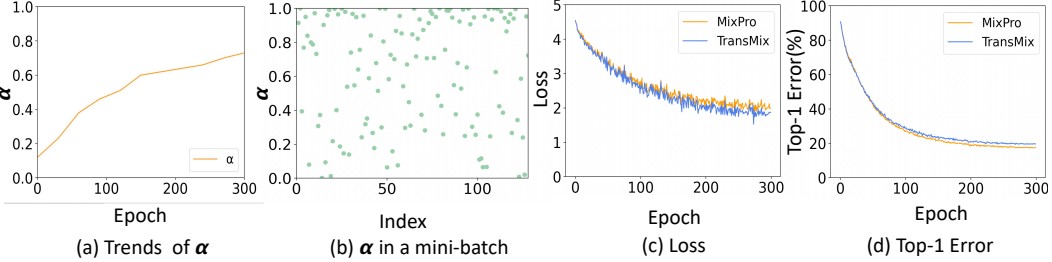

(a) Trends of $\boldsymbol{\alpha}$     (b) $\boldsymbol{\alpha}$ in a mini-batch     (c) Loss     (d) Top-1 Error

Figure 2: Visualization of progressive factor $(\alpha)$ and generalization analysis of our MixPro and TransMix. We employ Mixpro and TransMix based on Deit-S on ImageNet-1k for 300 epochs. Figures (a) and (b), demonstrate the visualization of the progressive factor $(\alpha)$. Figures (c) and (d) depict their loss and top-1 validation error on ImageNet-1k. MixPro provides improved generalization compared to TransMix.

**Visualization of progressive factor $(\alpha)$.** Fig. 2 (a) indicates the trend of the mean progressive factor $(\alpha)$ of all samples with the epoch. We can observe that since the model will gradually fit the training samples, the progressive factor will grow slowly with epoch. Fig. 2 (b) indicates the value of the progressive factor in a mini-batch at epoch 300. We can find that since the model learns to different degrees for different samples, the progressive factors obtained by the model will be different, so

our method can flexibly adjust the weight of the attention map in the mixed labels according to the progressive factors.

**Generalization analysis.** We compare and analyze the generalization of MixPro and TransMix. Fig. 2 (c) and (d) depict their loss and top-1 validation error on ImageNet-1k for 300 epochs based on Deit-S. We can notice that although MixPro has a larger loss in training compared to TransMix, it has a smaller top-1 error in verification. This demonstrates that Deit-S trained with MixPro can achieve better generalizability.

## 4 EXPERIMENT

In this section, we primarily evaluate MixPro for its effectiveness, transferability, and robustness. We first study the effectiveness of MixPro on ImageNet-1k classification in Sec. 4.1. Next, we show the transfer ability of a MixPro pre-trained model when it is fine-tuned for downstream tasks (. subTran). In the Sec.4.3, we also show that Mixpro can improve the model's robustness compared to TransMix.

### 4.1 IMAGENET CLASSIFICATION

Table 1: Compared to TransMix, MixPro provides better performance on a wide range of model variants, e.g. DeiT, PVT, CaiT, XCiT , Swin on ImageNet-1k classification. All the baselines are reported in TransMix (Chen et al., 2021).

| Models | Params | #FLOPs | Top-1 Acc(%) | Top-1 Acc(%) +TransMix | Top-1 Acc(%) +MixPro |
|---|---|---|---|---|---|
| DeiT-T (Touvron et al., 2021a) | 5.7M | 1.6G | 72.2 | 72.6 | 73.8+(+1.2) |
| PVT-T (Wang et al., 2021) | 13.2M | 1.9G | 75.1 | 75.5 | 76.7+(+1.2) |
| XCiT-T (Ali et al., 2021) | 12M | 2.3G | 79.4 | 80.1 | 81.2+(+1.1) |
| CA-Swin-T (Liu et al., 2021) | 28.3M | 4.2G | 81.6 | 81.8 | 82.8+(+1.0) |
| CaiT-XXS | 17.3M | 3.8G | 79.1 | 79.8 | 80.6+(+0.8) |
| DeiT-S (Touvron et al., 2021a) | 22.1M | 4.7G | 79.8 | 80.7 | 81.3+(+0.6) |
| PVT-S (Wang et al., 2021) | 24.5M | 3.8G | 79.8 | 80.5 | 81.2+(+0.7) |
| XCiT-S (Ali et al., 2021) | 26M | 4.8G | 82.0 | 82.3 | 82.9+(+0.6) |
| CA-Swin-S (Liu et al., 2021) | 49.6M | 8.5G | 82.8 | 83.2 | 83.7+(+0.5) |
| PVT-M (Wang et al., 2021) | 44.2M | 6.7G | 81.2 | 82.1 | 82.7+(+0.6) |
| PVT-L (Wang et al., 2021) | 61.4M | 9.8G | 81.7 | 82.4 | 82.9+(+0.5) |
| XCiT-M (Ali et al., 2021) | 84M | 16.2G | 82.7 | 83.4 | 84.1+(+0.7) |
| DeiT-B (Touvron et al., 2021a) | 86.6M | 17.6G | 81.8 | 82.4 | 82.9+(+0.5) |
| XCiT-L (Ali et al., 2021) | 189M | 36.1G | 82.9 | 83.8 | 84.7+(+0.9) |

**Implementation Details.** For image classification, we evaluate on ImageNet-1K (Deng et al., 2009), which contains 1.28M training images and 50K validation images from 1,000 classes. We examined various baseline vision Transformer models using in TranMix (Chen et al., 2021) including DeiT (Touvron et al., 2021a), PVT (Wang et al., 2021), CaiT (Touvron et al., 2021b), XCiT (Ali et al., 2021), CA-Swin (Liu et al., 2021; Chen et al., 2021). Specifically, CA-Swin replaces the last Swin (Liu et al., 2021) block with a classification attention (CA) block without parameter overhead, which makes it possible to generalize TransMix and MixPro onto Swin. The training schemes will be slightly adjusted to the official papers' implementation. These experiments' settings follow TransMix(Chen et al., 2021). We employ an AdamW optimizer for 300 epochs expect that XCiT (Ali et al., 2021) and CaiT (Touvron et al., 2021b) reported 400 epochs. Following the TransMix, we use a cosine decay learning rate scheduler and 20 epochs of linear warm-up. Similarly, a batch size of 1024, an initial learning rate of 0.001, and a weight decay of 0.05 are used. All baselines have already contained the carefully tuned regularization methods reported in TransMix (Chen et al., 2021) except for repeated augmentation (Hoffer et al., 2020) and EMA (Polyak & Juditsky, 1992) which do not enhance performance. Following TransMix(Chen et al., 2021), the attention map $\mathbf{A}$ in Eq. 5 can be obtained as an intermediate output from the multi-head self-attention layer of the last transformer block.

### 4.1.1 COMPARISON WITH TRANSMIX ON A WIDE RANGE OF VIT-BASED MODEL.

As shown in Table 1, MixPro can enhance the top-1 accuracy on ImageNet for all the listed ViT-based models. Compared to TransMix, MixPro achieves better performance on all models. In particular, MixPro has better performance on models with fewer parameters. For example, MixPro reaches 73.8% on Deit-T, which is 1.2% higher than TransMix.

### 4.1.2 COMPARISON WITH SOTA MIXUP VARIANTS

Table 2: Top1-accuracy, training speed (im/sec) and number of parameters comparison with state-of-the-art Mixup variants on ImageNet-1k for 300 epochs. For a fair comparison, all listed models are built upon DeiT-S training recipe. Training speed (im/sec) takes account of data mixup, and model forward and backward in train-time. The main results are reported in TransMix (Chen et al., 2021) and TokenMix (Liu et al., 2022).

| Method | Backbone | #Params | Speed (im/sec) | top-1 Acc (%) |
|---|---|---|---|---|
| Baseline | | 22M | 322 | 78.6 |
| GridMix | | 22M | 322 | 79.5 (+0.9) |
| CutMix | | 22M | 322 | 79.8 (+1.2) |
| Attentive-CutMix | DeiT-S | 46M | 239 | 77.5 (-1.1) |
| SaliencyMix | | 22M | 314 | 79.2 (+0.6) |
| Puzzle-Mix | | 22M | 139 | 79.8 (+1.2) |
| TransMix | | 22M | 322 | 80.7 (+2.1) |
| TokenMix | | 22M | 322 | 80.8 (+2.2) |
| MixPro(Ours) | | 22M | 322 | **81.3** (+2.7) |

We compare many SOTA Mixup-based methods on ImageNet-1k (Deng et al., 2009) in this section. Following TransMix, we also train based on DeiT-S for a fair comparison. MixPro is measured in image per second (im/sec), training speed (i.e., training throughput), and takes into account data mixup (Zhang et al., 2017), model forward and backward in train-time in an average of five runs for images at resolution 224x224 under 128 batch size with a TeslaV100 graphic card. Table 2 shows that MixPro outperforms all other Mixup-based methods.

### 4.2 TRANSFER TO DOWNSTREAM TASKS

We demonstrate the transferability of our MixPro-based pre-trained models to the downstream tasks, including semantic segmentation, object detection, and instance segmentation. We observe the enhancements over the vanilla pre-trained baselines and TransMix.

### 4.2.1 SEMANTIC SEGMENTATION

| pretrained | Backbone | Decoder | mIoU | +MS |
|---|---|---|---|---|
| ResNet101 | ResNet101 | Deeplabv3+ | 47.3 | 48.5 |
| DeiT-S | | | 49.1 | 49.6 |
| +TransMix | DeiT-S | Linear | 49.7 | 50.3 |
| +MixPro | | | **50.3** | **50.9** |
| DeiT-S | | | 49.7 | 50.5 |
| +TransMix | DeiT-S | Segmenter | 50.6 | 51.2 |
| +MixPro | | | **51.1** | **51.6** |

| Backbone | #Params | Object detection $AP^b$ | $AP^b_{50}$ | $AP^b_{75}$ | Instance segmentation $AP_m$ | $AP^m_{50}$ | $AP^m_{75}$ |
|---|---|---|---|---|---|---|---|
| ResNet50 | 44.2M | 38.0 | 58.6 | 41.4 | 34.4 | 57.1 | 36.7 |
| ResNet101 | 63.2M | 40.4 | 61.1 | 44.2 | 36.4 | 57.7 | 38.8 |
| PVT-S | 44.1M | 40.4 | 62.9 | 43.8 | 37.8 | 60.1 | 40.3 |
| TransMix-PVT-S | 44.1M | 40.9 | 63.8 | 44.0 | 38.4 | 60.7 | 41.3 |
| MixPro-PVT-S | 44.1M | **41.4** | **64.2** | **44.4** | **38.9** | **61.1** | **41.7** |

Table 3: Overhead-free impact of MixPro on transferring to a downstream semantic segmentation task on the Pascal Context (Mottaghi et al., 2014) dataset. (MS) denotes multiscale testing. The best results are in bold.

Table 4: Following TransMix (Chen et al., 2021), Overhead-free impact of MixPro on transferring to downstream object detection and instance segmentation using Mask R-CNN (He et al., 2017) with PVT (Wang et al., 2021) backbone on COCO val2017. $AP^b$ denotes bounding box AP for object detection, and $AP^{mk}$ denotes mask AP for instance segmentation.

**Settings.** We decode the sequence of patch encoding $\mathbf{z}_{patches} \in \mathbb{R}^{p \times d}$ to a segmentation map $\mathbf{s} \in \mathbb{R}^{H \times W \times K}$ in our experiments, where K is the number of semantic classes. Following TransMix (Chen et al., 2021), we also employ two convolution-free decoders. The first one is the linear decoder, which is a point-wise linear layer on DeiT. Patch encoding $\mathbf{z}_{patches} \in \mathbb{R}^{p \times d}$ is also employed to generate patch-level logits, which are reshaped and bilinearly upsampled to the segmentation map $\mathbf{s}$. The second one is the segmenter decoder (Strudel et al., 2021), which is a transformer-based decoder, namely the Mask Transformer. We also train and evaluate the models on the Pascal Context (Mottaghi et al., 2014) dataset, which contains 4998 images with 59 semantic classes and a background class. In addition, the reported mean Intersection over Union (mIoU) is averaged over all classes as the main metric. The experiments are carried out using MMSegmentation (Contributors, 2020). All comparative experimental results come from TransMix (Chen et al., 2021).

**Results.** Table 3 shows that the MixPro pre-trained DeiT-SLinear and DeiT-S-Segmenter outperform the TransMix pre-trained baselines of 0.6% and 0.5% mIoU, respectively. For multi-scale testing. MixPro also outperforms the TransMix pre-trained baselines of 0.6% and 0.4% mIoU.

### 4.2.2 OBJECT DETECTION AND INSTANCE SEGMENTATION

**Settings.** Object detection and instance segmentation experiments are conducted on COCO 2017, which contains 118K images and evaluates 5K validation images. Following TransMix, we employ our method on PVT (Wang et al., 2021) as the detection backbone since its pyramid features make it beneficial to object detection. Following TransMix (Chen et al., 2021), we adopt 1x training schedule (i.e., 12 epochs) to train the detector based on mmDetection (Chen et al., 2019) framework.

**Results.** As indicated in Table 4, we observe that on two downstream tasks, the results of MixPro's pre-trained backbone once again outperformed TransMix's pre-trained backbone, which enhanced 0.5% box AP and 0.5% mask AP, respectively.

## 4.3 ROBUSTNESS ANALYSIS

We also compare MixPro with TransMix in terms of robustness and out-of-distribution performance.

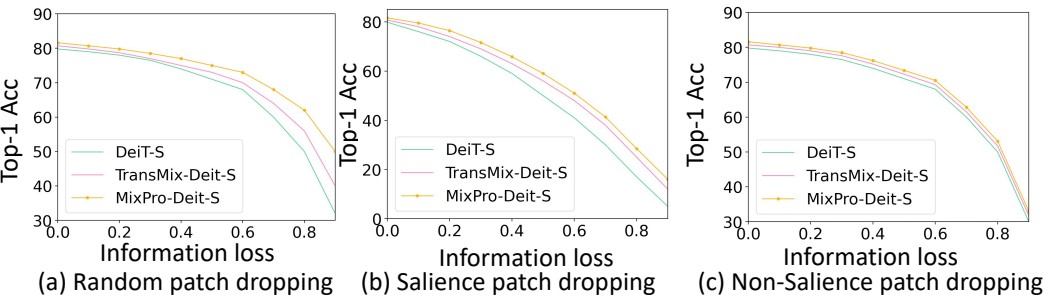

Figure 3: Robustness against occlusion. The figure shows the robustness of DeiT-S against occlusion with different information loss ratios.

### 4.3.1 ROBUSTNESS TO OCCLUSION

Naseer et al. (Naseer et al., 2021) investigate whether ViTs perform robustly in the presence of missing partial or substantial image content. In particular, ViTs divide an image into $N = 196$ patches on a 14x14 spatial grid. For example, an image of size 224×224×3 is split into 196 patches, each of size 16×16×3. In the experiments, "patch dropping" means replacing original image patches with 0-value patches. Following TransMix (Chen et al., 2021), we showcase the classification accuracy on an ImageNet-1k validation set with three dropping settings. (1) Random Patch Dropping, which selects and drops a subset of $M$ patches at random. (2) Salient (foreground) Patch Dropping, which studies the robustness of ViTs against occlusions of highly salient regions. Naseer et al. (Naseer et al., 2021) thresholds on DINO's attention map to obtain salient patches, which are dropped by ratios. (3) Non-salient (background) Patch Dropping, in which the least salient regions of an image are selected and dropped following the above approach.

**Results.** As demonstrated in Fig. 3, DeiT-S with MixPro is superior to TransMix and vanilla DeiT-S on all occlusion levels.

### 4.3.2 Natural Adversarial Example

Table 5: The robustness of DeiT-S against natural adversarial examples on ImageNet-A and out-of-distribution examples on ImageNet-O.

| Models | Nat. Adversarial Example | | | Out-of-Dist |
| | Top1-Acc | Calib-Error | AURRAC | AUPRC |
|---|---|---|---|---|
| DeiT-S | 19.1% | 32.0 % | 23.8 % | 20.9 % |
| TransMix-DeiT-S | 21.1% | 31.2 % | 28.8 % | 21.9 % |
| MixPro-DeiT-S | **22.4**% | **30.3** % | **32.4** % | **23.1** % |

For these experiments, we use the ImageNet-A dataset (Hendrycks et al., 2021), which adversarially collects 7,500 unmodified, natural but "hard" real-world images. They are drawn from some challenging scenarios ,such as fog scenes and occlusion. For evaluating our method, following settings in TransMix (Chen et al., 2021), we evaluate methods on the top-1 accuracy, Calibration Error (CalibError) (Hendrycks et al., 2021) which judges how classifiers can reliably forecast their accuracy, and the Area Under the Response Rate Accuracy Curve (AURRAC) which is an uncertainty estimation metric.

**Results.** As demonstrated in Table 5, MixPro-trained Deit-S outperforms TransMix-trained DeiT-S and vanilla DeiT-S on all metrics. MixPro lifts Top1-Acc on ImageNet-adversarial by 1.3%. For AURRAC, MixPro-DeiT-S achieves 32.4%, 3.6% higher than TransMix-DeiT-S.

### 4.3.3 Out-of-distribution Detection

The ImageNet-O (Hendrycks et al., 2021) is an adversarial out-of-distribution detection dataset. It adversarially collects 2000 images from outside ImageNet-1K. The anomalies of unforeseen classes should result in low-confidence predictions. The metric is the area under the precision-recall curve (AUPRC). Table 5 shows that MixPro-trained DeiT-S outperforms TransMix-trained DeiT-S by 1.2% AUPRC and outperforms DeiT-S by 2.2% AUPRC.

## 5 Conclusion

In this paper, we propose a new data augmentation method, MixPro. MixPro addresses the shortcomings of the current SOTA data augmentation method, TransMix, from the perspective of image and label space for the vision transformer. From the perspective of image space, we propose MaskMix which is a random mask strategy with adjustable scale. MaskMix ensures each image patch comes from only one image and contains more global contents. From the perspective of label space, we propose progressive attention labeling, which utilizes a progressive factor ($\alpha$) to dynamically re-weight the attention weight of the mixed attention label. Experimental results show that compared with TransMix, our method brings an improvement of 1.2%, 0.6%, and 0.5% for DeiT-T, DeiT-S, and Deit-B on the imagenet, respectively. After being pre-trained with MixPro on ImageNet, the ViT-based models also demonstrate better transferability to three downstream tasks such as semantic segmentation, object detection, and instance segmentation . In addition, compared to TransMix, MixPro also shows stronger robustness on three different benchmarks. In the ablation study, we detail the effect of each proposed module, the effect of different scales of mask patches, the different strategies of the progressive factor, and so on.

**Acknowledgments and Disclosure of Funding**

This work was supported in part by the National Natural Science Foundation of China under Grant 62271034, and in part by the Fundamental Research Funds for the Central Universities under Grant XK2020-03.

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

# A APPENDIX

## A.1 ABLATION STUDY

**The effect of our proposed module.** Table 6 demonstrated the effect of removing different components to provide insights into what makes MixPro successful. The experimental results are based on DeiT-S training for 300 epochs and Mixup (Zhang et al., 2017) is default used. We can observe that combining MaskMix and PAL, our method MixPro can enhance 1.5% compared to the standard setup (using Mixup and CutMix together), reaching 81.3% top-1 accuracy.

Table 6: Top1-accuracy is on imagenet-1K based on DeiT-S. We study the effect of removing different components to provide insights into what makes MixPro successful.

| CutMix | TransMix | MaskMix | PAL | top-1 Acc (%) |
|:---:|:---:|:---:|:---:|:---:|
| ✓ | ✗ | ✗ | ✗ | 79.8 |
| ✓ | ✓ | ✗ | ✗ | 80.7 (+0.9) |
| ✗ | ✓ | ✓ | ✗ | 81.0 (+1.2) |
| ✓ | ✓ | ✗ | ✓ | 81.1 (+1.3) |
| ✗ | ✗ | ✓ | ✓ | 81.3 (+1.5) |

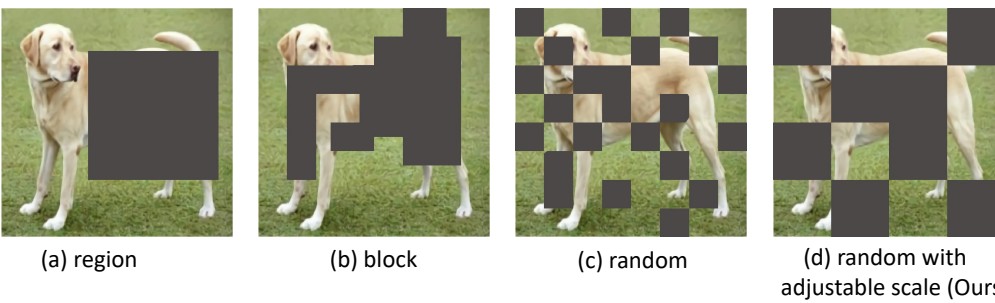

(a) region     (b) block     (c) random     (d) random with adjustable scale (Ours)

Figure 4: Illustration of different mask strategies.

**The effect of different mask strategies.** Figure 4 illustrates the different mask strategies. The region-based strategy is widely used in previous works (Yun et al., 2019; Uddin et al., 2020; DeVries & Taylor, 2017). Block and random based strategies mainly used in self-supervised learning (He et al., 2022; Bao et al., 2021). Our MixPro boosts random strategy on an adjustable scale. To better inspect the impact of mask strategies alone, in table 7, we directly use $\lambda_{area}$ to generate mixed labels instead of PAL. Table 7 shows the different performances of the mask strategies. Block and random strategy results come from TokenMix (Liu et al., 2022). We can observe that the block and random mask strategy improves significantly compared to the region mask strategy. This also indicates that images with more mixed global content are more effective for the vision transformer. Furthermore, our proposed adjustable scale is necessary for the random mask strategy. Compared with block and random strategies, the mask patches with a 4x scale can improve the accuracy of Deit-T by 0.5% and Deit-S by 0.2%.

Table 7: Ablation of mask strategy.

| model | region | block | random | random(4x scale) |
|---|---|---|---|---|
| Deit-T | 72.2 | 72.7 | 72.7 | **73.2** |
| Deit-S | 79.8 | 80.6 | 80.6 | **80.8** |

**The effect of different scales of mask patches.** The scale of mask patches $P_{mask}$ is multiple of the scale of image patches. There are several optional scales $\in \{1\times, 2\times, 4\times, 7\times\}$. The Fig. 5 (a) indicates that the evaluate results of MixPro with different scales based on DeiT-S on Imagenet-1K top-1 error. For all scales considered, MixPro improves upon the baseline (20.2%) significantly. Moreover, it performs best when the scale is multiplied by $4\times$. We still recommend adjusting the scale size more finely for different models to get better results.

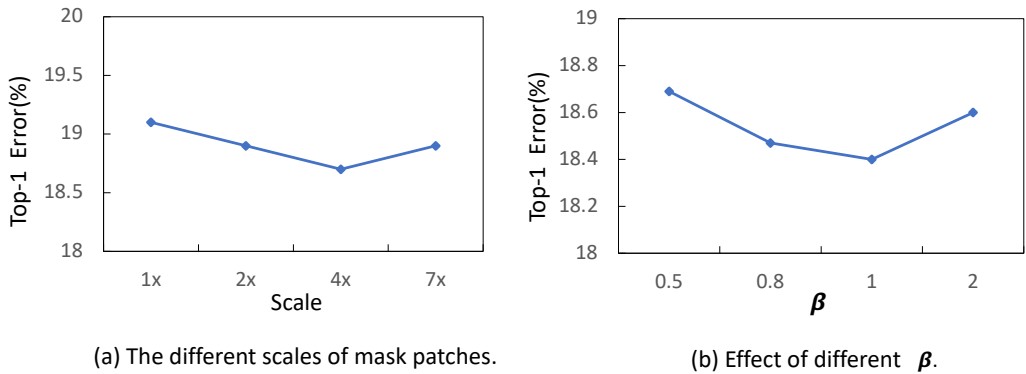

(a) The different scales of mask patches.

(b) Effect of different $\beta$.

Figure 5: Effect of different scales of mask patch and $\beta$ on Imagenet-1K top-1 error with DeiT-S.

**The effect of different $\beta$.** In Fig. 5 (b), we demonstrate the top-1 error of MixPro under different Beta distribution, such as $\beta \in \{0.5, 0.8, 1, 2\}$, on Imagenet-1K with DeiT-S. We can observe that $\beta$ is equal to 1 when our model achieves the best performance.

**The different strategies of progressive factor $\alpha$.** To facilitate the understanding of our proposed progressive attention labeling (PAL), we explore several different strategies to generate the progressive factor $\alpha$ evaluating on Imagenet-1K with DeiT-T. We employ several types of strategies: 1) Progress-relevant strategies adjust $\alpha$ with the number of training epochs, e.g., parabolic decay, etc. 2) Progress-irrelevant strategies include the equal weight. 3) A learnable parameter.

**How the boundary inside the patch influences ViTs?** TransMix introduces sharp rectangular borders even within an image patch (see Fig. 1, red colored) that are clearly distinguishable from the background. These borders inside the patch may draw excessive attention to the model, because for vision transformers, a patch would form a semantic unit, and the computation of the attention map of the whole image is obtained by summing the attention of each image patch.

Our MaskMix is able to eliminate the influence of the internal borders of patches. By aligning the borders of the mask patches with the image patches, there thus will not be sharp rectangular borders within image patches. Here also design three approaches to explore the performance of our MaskMix: **A. Aligning mask borders to patch borders for TransMix. B. Misaligning the borders of mask patches with image patches for our MaskMix. C. Reducing the scale of each mask unit for our MaskMix to introduce more noises.** Experimental results are below:

Table 8: Three approaches to exploring the performance of our MaskMix.

| method | TransMix | A | B | C | MaskMix(Ours) |
|---|---|---|---|---|---|
| Top-1 Acc (%) | 72.6 | 73.1 | 73.1 | 73.4 | **73.8** |

We can observe that our MaskMix introduces more noisy patterns (such as B and C) lead to worse results. The TransMix eliminates the internal borders of the patches using the approach A and lifting Top-1 Acc by 0.5%. This shows the improvements brought by the design in our MaskMix.

**Whether introducing more random patterns will further improve the performance?** We test three random patterns to evaluate the DeiT-T on ImageNet. The patterns and results are below:

Table 9: Evaluated on more noisy patterns.

| model | results |
|---|---|
| Baseline | 73.2 |
| Random shuffling on the cropped patches | 69.8 |
| Random shuffling on the patches of the canvas image | 69.7 |
| Both | 67.3 |

We can observe that damaging the original geometry of the image with these schemes leads to worse results, i.e., simply introducing random patterns (e.g., via simple random shuffling) does not bring performance improvement.

Table 10: Ablation studies of different progressive factor strategies on Imagenet-1K with on DeiT-T.

| strategy | $\alpha$ | Top-1 Error (%) |
|---|---|---|
| Equal weight | 0.5 | 27.3 |
| Linear increment | $\frac{T}{T_{max}}$ | 27.1 |
| Parabolic increment | $\frac{T}{T_{max}}2$ | 26.9 |
| Learnable parameter | - | 27.0 |
| PAL (Ours) | cosine distance | **26.2** |

As illustrated in Table 10, the progress-relevant strategies (i.e., linear increment, and parabolic increment) for generating $\alpha$ can yield better results than the progress-irrelevant strategy. The effects of progress-relevant strategies and learnable parameter strategy are equally. These observations prove our motivation that the model gets a low-confident attention map during the early training process, so employing the low-confident attention map for mixed labels may not be a good choice. Among these strategies, the one with the best performance for generating $\alpha$ is our proposed PAL.

## A.2 VISUALIZATION

As shown in Fig. 6, we visualize images generated by our method and its corresponding attention map. We can observe that mixed images produced by MaskMix consist of multiple scattered and continuous areas in the image, which thus can capture broader parts of the object content of the original images from a global perspective. Furthermore, the attention maps of these mixed images also make sense, capturing the feature of objects in mixed images.

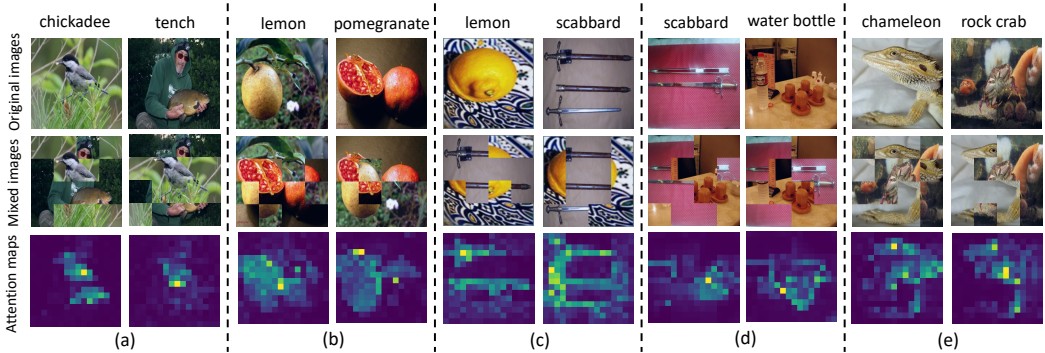

Figure 6: Visualization. Top: Original images. Medium: Mixed images generated by our MaskMix. Bottom: Attention maps of mixed images.

## A.3 TRAINING DETAILS ON IMAGENET-1K

Our training receipt follows previous works (Touvron et al., 2021a; Chen et al., 2021). The default setting is in Table 11.

Table 11: Training settings on ImageNet-1K.

| config | value |
|---|---|
| optimizer | AdamW |
| learning rate | 0.001 |
| weight decay | 0.05 |
| batch size | 1024 |
| learning rate schedule | cosine decay |
| warm-up epochs | 20 |
| training epochs | 300 |
| augmentation | RandAug(9, 0.5) |
| label smoothing | 0.1 |
| drop path | 0.1 |
| MixUp | 0.8 |
| CutMix | 1 |
| MixPro | 1 |

## A.4 PSEUDO-CODE

Algorithm 1 provides the pseudo-code of MixPro in a pytorch-like style. It demonstrates that simply few lines of code can boost the performance in the plug-and-play manner.

---

**Algorithm 1** Pseudo-code of MixPro in a PyTorch-like style.

```
# H, W: the height and width of the input image.
# h, w: the height and width of the attention map.
# M: 0/1 mask of MaskMix with shape (H,W).
# downsample: downsample from (H,W) to (h,w).

for (x, y) in dataloader: # load a mini-batch

    τ = Beta(β,β) # Eq. (6)
    M = mask_generate(τ, P_mask, P_image)
    λ_area = sum(M)
    x̃ = x * M + x.flip(0) * (1-M) # Eq. (7)
    logits, A = model(x̃)

    ỹ = λ_area * y + (1-λ_area) * y.flip(0) # Eq. (7)
    α = cos_similarity(logits, ỹ) # Eq. (8)

    M' = downsample(M)
    λ_attn = matmul(A, M') # Eq. (5)
    λ = α * λ_attn + (1-α) * λ_area # Eq. (9)
    ỹ = λ * y + (1-λ) * y.flip(0)

    CrossEntropyLoss(logits, ỹ).backward()
```

---

