# OpenReview forum: "MixPro: Data Augmentation with MaskMix and Progressive Attention Labeling for Vision Transformer"
_ICLR.cc/2023/Conference — ICLR 2023 poster_

### Official Review · Reviewer_ujft · 2022-10-20

**Confidence:** 5
**Correctness:** 4
**Technical Novelty And Significance:** 2
**Empirical Novelty And Significance:** 2
**Recommendation:** 5

**Clarity, Quality, Novelty And Reproducibility:**

As discussed above, the paper is of high quality and the method very clear. However originality is limited to incremental ideas and small improvements. The paper relies too heavily on TransMix.

**Strength And Weaknesses:**

The paper is well written and a pleasure to read. The ideas are very simple and the two improvements make sense. There are extensive experiments and comparisons, mostly following TransMix.

I find the two improvements incremental. They are both of the kind one would expect to find between a conference paper and a journal submission, not in a new paper.

The most critical missing experiment is an independent evaluation of the two components. As it stands, it is impossible to assess the benefit of either idea (MaskMix and PAL). There is an ablation in Table 6 (Appendix) but only on ImageNet-1k accuracy, where the gains of the two ideas are only 0.3% and 0.4%. Independent evaluation should follow on all experiments.

Another limitation of the empirical study is the narrow focus on vision transformers. TransMix may have initiated this but there are many ways one may obtain an attention map from a convolutional network, so I see no reason for this narrow focus. Extending to convolutional networks will also allow a comparison with previously published results. Currently, the only comparison with methods prior to TransMix is on ImageNet-1k classification (Table 2). All other comparisons are limited to TransMix and the baseline, which is not sufficient.

MaskMix is motivated by the global receptive field of vision transformers, but I find this motivation insufficient. For one thing, region-level crops appear to be more difficult for the local receptive field of convolutional networks; for another, an independent evaluation of MaskMix and a study on convolutional networks could at least validate the claim.

Concerning PAL:
- The use of cosine similarity in (8) is not elegant or justified, since the two probability vectors are l1-normalized. More appropriate is the histogram intersection (which is to l1 norm what is dot product to l2 norm). Or any other similarity measure based on a proper distribution metric.
- The probability vector p of the interpolated input is not a good indicator of classifier confidence. More reliable are the probability vectors of the clean images, which could be compared with the clean labels (resulting in the predicted probability of the ground truth classes). I am not sure how this could be implemented, given that TransMix only processes mixed images. Maybe confidence could be estimated globally by considering some clean images per epoch. This could be at least a baseline for Table 8, even if it does not improve.
- Since patches are chosen at random from one or the other image, it is not clear if the attention map of a mixed image makes as much sense as that of a clean image or at least that of TransMix (CutMix) style mixing (by rectangular area). At least some visualization is in order. Another baseline is using always the area-based label interpolation as in CutMix.
- Of course, PAL should be evaluated independently of MaskMix (with rectangular area image mixing).

Another idea or baseline that relates to both improvements is to use attention maps not only for label but also for input image interpolation, as in SaliencyMix.

Other benchmarks that were considered in prior works and not in TransMix or this work include robustness to adversarial attacks, calibration and out-of-distribution detection on datasets other than ImageNet, where the network is trained. Such experiments are important not just for completeness but because they have margin for considerable gains for a new method. For example, the gain of MixPro over TransMix is in the order of 0.5% in most experiments. Given that MixPro consists of two ideas, the gain of each is even less.

**Summary Of The Paper:**

This work proposes MixPro, an improvement of TransMix, a mixup method designed for vision transformers where the label interpolation weights are determined by the attention maps. It contributes two ideas: 1) (MaskMix) in the input space, the mask is random over patches rather than rectangular (as in CutMix). 2) (Progressing Attention Labeling, PAL) The label interpolation weights evolve from area-based (as in CutMix) to attention-based (as in TransMix) as the confidence of the classifier increases during training, thus attention maps becoming more reliable.

**Summary Of The Review:**

Overall, this work proposes two simple and interesting ideas and is well implemented and well written, but given its limited novelty and limited benefit, one would expect as compensation at least a wider range of empirical studies that would allow a deeper understanding of the related methods. The current study is not mature enough and I am not sure if it can make it for this ICLR.

---

> ### Author Response · Authors · 2022-11-19
> **Part 1 of Response to Reviewer ujft**
>
>
> Thanks for your detailed and constructive comments. We address the comments below:
>
> **Q1: I find the two improvements incremental. They are both of the kind one would expect to find between a conference paper and a journal submission, not in a new paper.**
>
> **A1:** The emergence of vision transformers (ViT) further improves the effectiveness of various visual tasks, which also poses the requirements of stronger data augmentation technologies. Previous data augmentation methods are mainly designed for convolutional networks, but little work has specifically focused on vision transformer which is also an important research direction. Our proposed MixPro can significantly improve the performance of ViT-based methods with our newly designed method with more global friendly mixed images and the new scheme for label assignment. Our results show the efficacy of such a method, which could benefit the performance improvement of transformers on handling various visual tasks.
>
> **Q2: The most critical missing experiment is an independent evaluation of the two components. As it stands, it is impossible to assess the benefit of either idea (MaskMix and PAL). There is an ablation in Table 6 (Appendix) but only on ImageNet-1k accuracy, where the gains of the two ideas are only 0.3\% and 0.4\%. Independent evaluation should follow on all experiments.**
>
> **A2:** Thanks for the suggestion. We emphasize that these systematic improvement using data augmentation on ImageNet-1k is still significant when compared to the methods with structure modification on models. For example, the engineering new architectures such as PiT-B [1] and CrossViT-B [2] only lift the DeiT-B baseline result by 0.2\% and 0.4\%, respectively. Following reviewers' comments, we also conduct independent evaluation on other experiments, results are below. We can observe that eitheridea brings consistent performance improvement.
>
> * ImageNet Classification.
> |model |baseline|+TransMix|+our MaskMix only|+our PAL only|
> |-|--:|--:|--:|--:|
> |DeiT-T    | 72.2 |  72.6  |73.4 |73.5 |
> |PVT-T     | 75.1 |  75.5  |76.4 |76.3 |
> |XCiT-T    | 79.4 |  80.1  |81.8 |81.9 |
> |CaiT-XXS  | 79.1 |  79.8  |80.4 |80.5 |
> |DeiT-S    | 79.8 |  80.7  |81.0 |81.1 |
> |DeiT-B    | 81.8 |  82.4  |82.7 |82.8 |
> |PVT-M     | 81.2 |  82.1  |82.5 |82.5 |
> |XCiT-L    | 82.9 |  83.8  |84.3 |84.4 |
>
> * Semantic segmentation. Evaluate on mIou.
> |decoder |baseline|+our MaskMix only|+our PAL only|
> |-|--:|--:|--:|
> |Linear | 49.1 |  50.1 |  50.1 |
> |Segmenter | 49.7 |     50.9 |  51.0 |
>
> * Object detection .
> |decoder |PVT-T|+our MaskMix only|+our PAL only|
> |-|--:|--:|--:|
> |AP | 40.4 | 41.2 |  41.1|
>
> * Instance segmentation
> |decoder |PVT-T|+our MaskMix only|+our PAL only|
> |-|--:|--:|--:|
> |AP | 37.8 |  38.6 |  38.8 |
>
> We can observe that the independent evaluation of our MaskMix and PAL also bring significant improvement on all experiments.

---

> ### Author Response · Authors · 2022-11-19
> **Part 2 of Response to Reviewer ujft**
>
>
> **Q3: Another limitation of the empirical study is the narrow focus on vision transformers. TransMix may have initiated this but there are many ways one may obtain an attention map from a convolutional network, so I see no reason for this narrow focus.**
>
> **A3:** Thanks for the suggestion. Recently, some data augmentation methods were inspired by Mixup [9] for regularizing convolutional network, yet little work has specifically focused on vision transformers which yet is also an important research direction. This is one of the reasons that motivated us to design a data augmentation method for visual transformer characteristics. Our MixPro can also be adapted and employed for convolutional networks, in which we use Grad-CAM [10] to get the attention maps. We then compare the state-of-the-art data augmentation methods including CutMix [3], Manifold [4], PuzzleMix [5], SaliencyMix [6], StyleMix [7] and AlignMixup [8] on some convolutional network visual tasks. We follow the setting of the prior art [8] to conduct the experiments, and the results are below:
>
> * Image classification top-1 error (\%) and computational analysis on ImageNet using  PreActResnet18 (R-18).
> |method |R-18|+CutMix|+Manifold|+PuzzleMix|+SaliencyMix|StyleMix|+StyleCutMix|+MixPro(ours)|
> |-|--:|--:|--:|--:|--:|--:|--:| --:|
> |CIFAR-100     | 23.24 |  19.37 |19.80 |20.01 |  19.69 |20.04 |19.34 |$\textbf{18.78}$ |
> |TinyInagenet  | 43.40 |  43.11 |40.76 |36.52 |  34.81 |36.13 |34.49 |$\textbf{34.02}$ |
>
> * Image classification top-1 error (\%) and computational analysis on ImageNet using WRN16-8 (W16-8).
> |method |W16-8|+CutMix|+Manifold|+PuzzleMix|+SaliencyMix|+StyleMix|+StyleCutMix|+MixPro (ours)|
> |-|--:|--:|--:|--:|--:|--:|--:|  --:|
> |CIFAR-100 |20.63 | 19.71 |19.23 |19.25 | 19.57 |20.45 |19.28 |$\textbf{18.67}$|
>
> * Image classification top-1 error (\%) and computational analysis on ImageNet using Resnet-50 (R-50).
> |method |R-50|+CutMix|+Manifold|+PuzzleMix|+SaliencyMix|+StyleMix|+AlignMixup|+MixPro(ours)|
> |-|--:|--:|--:|--:|--:|--:|--:|  --:|
> |ImageNet | 23.68 |  20.70 |22.50 |21.24 |  21.26 |- |20.68 |$\textbf{20.21}$|
>
> * Weakly-supervised object localization Top-1 localization accuracy (\%) on CUB200-2011.
> |method |Baseline|+CutMix|+MixPro (ours)|
> |-|--:|--:|--:|
> |CUB200| 49.4 |  54.8 |$\textbf{55.3}$|
>
> We can observe that the evaluation of our MixPro achieves significant improvement on the above visual tasks.
>
>
> **Q4: MaskMix is motivated by the global receptive field of vision transformers, but I find this motivation insufficient. For one thing, region-level crops appear to be more difficult for the local receptive field of convolutional networks. For another, an independent evaluation of MaskMix and a study on convolutional networks could at least validate the claim.**
>
> **A4:** Our MaskMix mixing two images mainly aims to provide data augmentation and regularization for vision transformers. It can also benefit convolutional networks, but has less gain compared to utilizing vision transformers as vision transformers have global receptive field. We evaluate our MaskMix on convolutional networks and DeiT, and the results are below:
> |model |region-level|MaskMix (ours)|
> |-|--:|--:|
> |DeiT-T| 72.5 |  73.2 |
> |DeiT-S| 79.8 |  80.7 |
> |ResNet-50| 79.3 |  79.8|
>
> Compared to convolutional networks, the MaskMix is more suitable for vision transformers with global receptive field and provides greater gains.
>
> **Q5: The use of cosine similarity in (8) is not elegant or justified, since the two probability vectors are l1-normalized. More appropriate is the histogram intersection (which is to l1 norm what is dot product to l2 norm). Or any other similarity measure based on a proper distribution metric.**
>
> **A5:** We utilize cosine similarity, because for two l1-normalized vectors, the results of cosine similarity is ranged to (0,1). The value thus can be used to trade-off the proportion of $\lambda_{attn}$ and $\lambda_{area}$. Other measures (such as histogram intersection) can also be used, but we may need to additionally design the function to map the final result to the range (0, 1). We also evaluated different similarity measure methods based on DeiT on ImageNet-1K, and the results are below:
>
> |method |Cosine similarity (ours)|Histogram intersection|Euclidean Distance|Jensen-Shannon Divergence|Cross Entropy|
> |-----|----|----|----|----|----|
> |Accuracy| $\textbf{73.2}$ |  72.8 |72.7 |72.9 |72.8|
>
> We observe that our PAL using other measure methods for Eq. (8) will result in a slight performance degradation.

---

> ### Author Response · Authors · 2022-11-19
> **Part 3 of Response to Reviewer ujft**
>
>
> **Q6: The probability vector p of the interpolated input is not a good indicator of classifier confidence. More reliable are the probability vectors of the clean images, which could be compared with the clean labels (resulting in the predicted probability of the ground truth classes). I am not sure how this could be implemented, given that TransMix only processes mixed images. Maybe confidence could be estimated globally by considering some clean images per epoch. This could be at least a baseline for Table 8, even if it does not improve.**
>
> **A6:** Thanks for the suggestion. We follow the suggestion to estimate a confidence threshold by jointing unmixed images for each mini-batch and considering clean images per epoch. We evaluate the approach as approach A, and the results are below:
>
> |method |A|+MixPro (ours)|
> |-|--:|--:|
> |Top-1 Error| 26.4|  26.2 |
>
> Compared with A, our MixPro can achieve better performance because of its dynamic progressive factor for label assignment.
>
> **Q7: Since patches are chosen at random from one or the other image, it is not clear if the attention map of a mixed image makes as much sense as that of a clean image or at least that of TransMix (CutMix) style mixing (by rectangular area). At least some visualization is in order.**
>
> **A7:** Thanks for the suggestion. We have added the visualization of the attention map and mixed image generated by our PAL and our MaskMix in Appendix.
>
> **Q8: Another baseline is using always the area-based label interpolation as in CutMix. Of course, PAL should be evaluated independently of MaskMix (with rectangular area image mixing).**
>
> **A8:** Thanks for the suggestion. Experimental results are below:
>
> * Evaluating our MaskMix with area-based label interpolation, and a baseline is using CutMix:
> |model |baseline|+MaskMix (ours)|
> |-|--:|--:|
> |DeiT-T | 72.2 |  73.2 |
> |DeiT-S | 79.8 |  80.7 |
> |DeiT-B | 81.8 |  82.3 |
> |XCiT-L | 82.9 |  83.7 |
>
> Compare with baseline, our MaskMix can achieve better performance on several models.
>
> * Evaluating our PAL with rectangular area image mixing , and a baseline is using CutMix:
> |model |baseline|+PAL (ours)|
> |-|--:|--:|
> |DeiT-T | 72.2 |  73.1 |
> |DeiT-S | 79.8 |  81.1 |
> |DeiT-B | 81.8 |  82.7 |
> |XCiT-L | 82.9 |  84.2 |
>
> Compare with baseline, our MaskMix can achieve better performance on several models.
>
>
> **Q9: Another idea or baseline that relates to both improvements is to use attention maps not only for label but also for input image interpolation, as in SaliencyMix.**
>
> **A9:** Thanks for the suggestion. We use attention maps to guide the image interpolation, and the results are below:
>
> * Evaluating on ImageNet-1K, and a baseline is using CutMix:
> |model |baseline|MaskMix (ours) +attention|
> |-|--:|--:|
> |DeiT-T | 72.2 |  73.6 |
> |DeiT-S | 79.8 |  81.2 |
> |DeiT-B | 81.8 |  82.7 |
> |XCiT-L | 82.9 |  83.9 |
>
> Compare with baseline, our MaskMix wich attention can also achieve better performance on several models.

---

> ### Author Response · Authors · 2022-11-19
> **Part 4 of Response to Reviewer ujft**
>
>
> **Q10: Other benchmarks that were considered in prior works and not in TransMix or this work include robustness to adversarial attacks, calibration and out-of-distribution detection on datasets other than ImageNet, where the network is trained.**
>
>
> **A10:** Thanks for the suggestion. We conduct the experiments and compare with state-of-the art (SOTA) methods based on convolutional neural network (CNN) including robustness to adversarial attacks, calibration, and out-of-distribution detection. We follow the setting and results of prior arts [6] to conduct these experiments, and our results are below:
>
> * Out-of-distribution detection for on LSUN and TinyImageNet (TI).
> Det Acc (detection accuracy),AuROC, AuPR (ID) and AuPR (OOD):higher is better
> |dataset |  |  LSUN (crop) |   |   |  |TI (crop)  |   |  |
> |-|--:|--:|--:|--:|-------:|--:|--:| --:|
> |method |Det Acc|AuROC|AuPR(ID)|AuPR(OOD)|Det Acc|AuROC|AuPR(ID)|AuPR(OOD)|
> |baseline|            54.0 |47.1 |54.5| 45.6 |61.2 |64.8 |67.8 |60.6|
> |+CutMix|             63.8 |63.1 |61.9| 63.4| 70.4 |84.3 |87.1 |80.6|
> |+Manifold|           58.9 |60.3 |57.8| 59.5 |67.4 |69.9 |69.3 |70.5|
> |+PuzzleMix|          64.3 |69.1 |80.6| 73.7 |71.8 |76.2 |78.2 |81.9|
> |+StyleMix|           62.3 |64.2 |70.9| 63.9 |67.8 |73.9 |71.5 |78.4|
> |+SaliencyMix|        68.5 |79.7 |82.2| 64.4 |73.3 |83.7 |87.0 |82.0|
> |+MixPro (ours)|      69.8 |80.3 |83.2| 74.3 |75.9 |84.6 |87.3 |83.1|
>
>
> * Calibration using PreActResnet18 on CIFAR-100.
> ECE: expected calibration error; OE: overconfidence error. Lower is better.
> |method |ECE|OE|
> |-|--:|--:|
> |baseline|     10.25| 1.11|
> |+CutMix|      7.60 |1.05|
> |+Manifold|    18.41| 0.79|
> |+PuzzleMix|   8.22 |0.61|
> |+StyleMix|    11.43| 1.31|
> |+SaliencyMix| 5.89 |0.59|
> |+MixPro (ours)|   6.2 |  0.58 |
>
> * Robustness to FGSM \& PGD attacks.
> Evaluate on Top-1 error (\%): lower is better.
> |attack|FGSM|FGSM|FGSM|PGD|PGD|
> |-|--|--| --| --| --|
> |dataset|      CIFAR-100|CIFAR-100|TinyImagenet|CIFAR-100|CIFAR-100|
> |method  |       ResNet-18|  W16-8|  R-18| ResNet-18|  W16-8|
> |baseline |       87.12 |72.81|91.85 |99.97 |99.99|
> |+CutMix  |       86.96 |60.16 |88.68| 98.67| 97.98|
> |+Manifold  |     80.29 |56.45 |89.25 |99.66 |98.43|
> |+PuzzleMix  |     78.70| 57.77 |83.91| 96.42 |95.28 |
> |+StyleMix    |    80.54| 67.94| 84.93 |98.39| 98.24|
> |+SaliencyMix  |   77.79 |58.10| 81.16|95.68 |93.76 |
> |+MixPro (ours)|  76.12|56.21|80.54|93.46|93.14|
>
> We can observe that the evaluation of our MixPro achieves significant improvement on the above tasks, including robustness to adversarial attacks, calibration, and out-of-distribution detection with different backbones and datasets.
>
>
> **Reference**
>
> [1] Byeongho Heo, Sangdoo Yun, Dongyoon Han, Sanghyuk Chun, Junsuk Choe, and Seong Joon Oh. Rethinking spa- tial dimensions of vision transformers. In Proceedings of the IEEE/CVF International Conference on Computer Vision (ICCV), 2021
>
> [2] Chun-Fu Chen, Quanfu Fan, and Rameswar Panda. Crossvit: Cross-attention multi-scale vision transformer for image classification. In Proceedings of the IEEE/CVF International Conference on Computer Vision (ICCV), 2021
>
> [3] Sangdoo Yun, Dongyoon Han, Seong Joon Oh, Sanghyuk Chun, Junsuk Choe, and Youngjoon Yoo. Cutmix: Regu- larization strategy to train strong classifiers with localizable features. In ICCV, 2019
>
> [4] Vikas Verma, Alex Lamb, Christopher Beckham, Amir Na- jafi, Ioannis Mitliagkas, David Lopez-Paz, and Yoshua Ben- gio. Manifold mixup: Better representations by interpolating hidden states. In ICML, 2019
>
> [5] Jang-Hyun Kim, Wonho Choo, and Hyun Oh Song. Puz- zle mix: Exploiting saliency and local statistics for optimal mixup. In ICML, 2020
>
> [6] A F M Uddin, Mst. Monira, Wheemyung Shin, TaeChoong Chung, and Sung-Ho Bae. SaliencyMix: A saliency guided data augmentation strategy for better regularization. In ICML, 2021
>
> [7] MinuiHong,JinwooChoi,andGunheeKim.Stylemix:Sep- arating content and style for enhanced data augmentation. In CVPR, 2021
>
> [8] Venkataramanan S, Kijak E, Amsaleg L, et al. AlignMixup: Improving Representations By Interpolating Aligned Features[C]//Proceedings of the IEEE/CVF Conference on Computer Vision and Pattern Recognition. 2022: 19174-19183.
>
> [9] Zhang H, Cisse M, Dauphin Y N, et al. mixup: Beyond empirical risk minimization[J]. arXiv preprint arXiv:1710.09412, 2017.
>
> [10] Selvaraju R R, Cogswell M, Das A, et al. Grad-cam: Visual explanations from deep networks via gradient-based localization[C]//Proceedings of the IEEE international conference on computer vision. 2017: 618-626.

---

> > ### Comment · Reviewer_ujft · 2022-11-23
> > **Thanks for the detailed responses**
> >
> > The authors have provided an impressive amount of new material in support of their method. This addresses my concerns to a large extent and improves the paper in several aspects.
> >
> > I am afraid that my concern on novelty remains and for this I have to keep my original recommendation. However, acknowledging that the other reviewers give positive recommendations, I would have no objection in accepting the paper.
> >
> > A further point that needs to be addressed is that the performance on the three additional tasks given in response part 4 does not appear to be competitive with AlignMixup [8], although ImageNet ResNet-50 classification accuracy in response part 2 is competitive. This highlights the importance of validating new ideas in a broader experimental setting, in terms of tasks as well as architectures. I wonder how [8] would perform on the transformer models studied here. This would give a more complete picture.

---

> > > ### Author Response · Authors · 2022-12-10
> > > **Response to Reviewer ujft**
> > >
> > > Thank you a lot for your valuable feedback, and pointing out that we have addressed your concerns to a large extent.
> > >
> > > In recent years, ViT-based methods have achieved state-of-the-art performance on various tasks, and are becoming main-stream architectures for different tasks. Different from other architectures (e.g., CNNs), where different mix-up methods have been designed, designing mix-up (augmentation) methods for ViT is much less explored. Yet, how to design mix-up (augmentation) methods to effectively enhance the performance of such an important architecture (ViT) is a very important problem.
> > >
> > > Considering the different properties of ViT compared to other networks (e.g., CNNs), in our paper, we propose a new method, MixPro for ViT, which specifically introduces more global-friendly patch-level mixed images and a new label assignment scheme. Our method specifically addresses the issues (e.g., border alignment and global-friendly design) when doing mix-up over the ViT token patches. Such a method is lightweight and can be easily implemented in ViT, yet due to our new design, it achieves state-of-the-art performance when added to ViT, and consistently benefits transformer performance improvement in various visual tasks. The consistent performance improvement suggests the efficacy of designing such ViT-more-friendly mix-up methods for handling ViT, and shows our contributions and novelties.
> > >
> > > Also note that, our designs (including border alignment and global-friendly design) are specifically designed for enhancing ViT, considering the importance of ViT architectures in different tasks. Thus, it is not specifically designed for comparison with AlignMixup [8] on CNNs (in response part 4).
> > >
> > > Here, following the reviewer’s suggestion, we also evaluate AlignMixup [8], besides the baseline Mixup, for Vision Transformers on the ImageNet-1k, and the results of comparisons with our MixPro are shown below:
> > >
> > > |model |baseline|+AlignMixup|+MixPro (ours)|
> > > |-|--:|--:|--:|
> > > |DeiT-T | 72.2 | 72.8 |  73.8 |
> > > |DeiT-S | 79.8 | 80.4 |  81.3 |
> > > |DeiT-B | 81.8 | 82.2 |  82.9 |
> > >
> > > Though AlignMixup is quite effective for CNN architectures, our above experimental results show that AlignMixup brings smaller improvements for ViT. However, benefiting from our delicate design of MixPro for Vision Transformers, our method is significantly better than AlignMixup in various architectural settings. This clearly shows the advantages of our new method with a ViT-friendly design for effectively enhancing ViT-based approaches.

---

> > > > ### Author Response · Authors · 2022-12-12
> > > > **Reply to Reviewer ujft**
> > > >
> > > > Dear Reviewer ujft,
> > > >
> > > > thank you a lot for your constructive and helpful comments. Could you help to check if our reply has addressed your concerns? Thank you!
> > > >
> > > > Best Regards
> > > >
> > > > Authors

---

> > > > ### Comment · Reviewer_ujft · 2022-12-12
> > > > **Thank you**
> > > >
> > > > I thank the authors for the additional results, which certainly make the picture more complete on ImageNet classification. My point above was referring to the experiments of part 4, so the picture would be even more complete if those experiments were performed using transformer models. Then one could indeed conclude how the proposed MixPro performs relative to the state of the art on a number of tasks, using convolutional or transformer models. This would be sufficient support for the high-level claims of the paper.
> > > >
> > > > Coming to the high-level claims, the authors repeat here a lengthy discussion on the model properties, their motivation and justification of the results. I am afraid this discussion does not bring new information comparing with the paper. To me, "global-friendly" remains an abstract, non-concrete, undefined concept and I still cannot see how "patch-level mixed images" are "global-friendly." This concern was expressed in my original review (paragraph "MaskMix is motivated ... the claim.") But let me make this concern more concrete here, with an example of what I meant "region-level crops appear to be more difficult for the local receptive field of convolutional networks."
> > > >
> > > > "DropBlock: A regularization method for convolutional networks" by Ghiasi et al. NeurIPS 2018 shows that dropping rectangular regions from convolutional feature maps is a more effective regularizer than dropping patches at random. Although this is a different problem than mixup, the high-level message is the opposite of the one given for this paper and the underlying explanation holds in all convolutional networks, regardless of the problem: "nearby activations contain closely related information." I hope it is clear now how "global friendly" still troubles me.
> > > >
> > > > My final recommendation remains as I explained in my previous response, meaning that, although negative, I am fine with accepting the paper according to the other two reviews. I hope this discussion helps in improving the paper in case it is accepted.

---

> > > > > ### Author Response · Authors · 2022-12-12
> > > > > **Response to Reviewer ujft**
> > > > >
> > > > > Thanks for your further detailed and knowledgeable comments. In our paper, we follow experiment settings and tasks of the previous work of TransMix in all the experiments, which is the state-of-the-art augmentation method for ViT. These experiments include not only mainstream vision tasks (classification, semantic segmentation, object detection, and instance segmentation), but also some robustness analysis and out-of-distribution detection. Our MixPro achieves consistently better performance than the previous state-of-the-art TransMix following all its  experimental settings. Thanks for the suggestion, we will also include the suggested settings in paper.
> > > > >
> > > > > For the "patch-level," "patch" refers to the patch of the mask, which is a multiple of the size of the ViT’s image patch (16 x 16). The area of the patch is thus not much smaller than the area of the region. So for our MaskMix, it is equivalent to having multiple regions distributed in different positions of the original image, while the region-based methods only have only one single region that can be mixed. The "global" means that, because the mask has multiple exchanged regions, it then have a higher probability of capturing different parts of the object contents as there are multiple patches distributed over different positions of the image. This leads to a more global-level augmentation.
> > > > >
> > > > > The high-level information in DropBlock is not the opposite of what is expressed in our paper. DropBlock also utilizes two or more drop blocks rather than only one drop block. In our paper, although it is a very different problem, MaskMix also utilizes multiple block regions to exchange image information instead of one as region-based. In Appendix A.2 of our paper, the scale of the mask patches in MaskMix is also important: 4x scale of $P_{mask}$ is better than 1x scale of $P_{mask}$. This can also match DropBlock’s view that "nearby activations contain closely related information."

---

### Official Review · Reviewer_vD1g · 2022-10-21

**Confidence:** 5
**Correctness:** 3
**Technical Novelty And Significance:** 3
**Empirical Novelty And Significance:** 4
**Recommendation:** 6

**Clarity, Quality, Novelty And Reproducibility:**

I recognize the novelty and the efficacy of the proposed method. The paper is basically clear but lacks some implementation details. The reproducibility cannot be guaranteed since the authors do not promise to make their code publicly available.

**Strength And Weaknesses:**

### Strength

**1. The proposed method is simple yet effective, which makes it simple to follow.** MixPro basically follows the paradigm designed in TransMix, which only requires several lines of code to implement. Personally I like ideas that are simple and sweet. The major changes in MixPro, including MaskMix and Progressive Attention Labeling, are easy to be implemented in principle. However, the authors did not include the pseudo-code in their manuscript. It would be more clear if they can include the pseudo-code in the appendix.

**2. The proposed method exhibit strong and consistent improvement** for different tasks and models even when compared with a strong baseline TransMix.

**3. The proposed method is fast and applicable to many different Transformer-based networks.** As shown in Table 2, the proposed method can lead to a significant gain with minimal computational introduced.

**4. The ablation studies and experiments are comprehensive and extensive.** The comparison between MixPro and TransMix makes it clear to understand the efficacy of the proposed method.

### Weakness

**1. I am not fully persuaded by the whole story and the explanation given by the author in the introduction.** The authors argue that TransMix has three major drawbacks including region-based mixing, boundary artifact and unreliable attention matrix.
- 1.1 For region-based mixing, I am a bit confused about why *region-based mixed images may provide insufficient image contents* and why the proposed MaskMix can help to alleviate this problem as MaskMix will also mix just a portion of the entire image, which may be insufficient as well.
- 1.2 For boundary artifact, the authors argue that it will introduce *sharp rectangular borders that
are clearly distinguishable from the background.* However, the solution MaskMix will also lead to these sharp artifacts and the patch-wise mixture will spread this kind of noise everywhere in the mixed image, which may even aggravate such problem. Instead of echoing the argument by the authors, I would suggest that MaskMix may introduce more noisy patterns into the image mixture so that the network can be better regularized by these noises.
- 1.3 For Progressive Attention Labeling (PAL), the authors argue that it is better than TransMix since *attention maps may not always be reliable during the training process.* However, the $\lambda$ of TransMix is actually the mean of attention weights and the original $\lambda$ introduced by CutMix (a.k.a. $\lambda_{area}$ in the manuscript). This indicates that TransMix also considers this problem and uses the area of the cropped image as an extra condition. Meanwhile, TransMix also evaluates the effect of using different attention maps in calculating $\lambda_{attn}$. However, using a highly-discriminative attention map pre-trained by DINO to calculate $\lambda_{attn}$ does not result in any improvement in terms of the top-1 accuracy on ImageNet (see in Table 6 in TransMix paper). So I would doubt whether the label noise introduced in the label space would also be helpful.

**2. The illustration for details of the proposed method is a bit unclear to me.** The authors did a good job in describing their overall pipeline and the basic idea. However, I am a bit confused about whether the introduced image will be first shuffled or not before mixing. The cat in Figure 2 does not follow the original geometric structure, but there seems to be no shuffling operation in Eq. (7).

**3. May the authors do more analysis on why and how MaskMix works better than TransMix?** My assumption is that **MixPro creates more complex scenarios in the image mixture such that the network can be better regularized**.  Thereby I am curious about whether introducing more random patterns will further improve the performance. These random patterns may include but not be limited to:
- random shuffling (or shifting) on the cropped patches
- random shuffling (or shifting) on the patches of the canvas image

**4. The proposed method is only applicable to Vision Transformers with class token.**

**Summary Of The Paper:**

This paper presents a new variant of CutMix, namely MixPro to further improve the previous state-of-the-art (TransMix) for Vision Transformers. The proposed method, including MaskMix and Progressive Attention Labeling, aims to tackle three primary drawbacks they found in TransMix. Extensive experiments show that MixPro can lead to consistent and remarkable improvement on various tasks and datasets.

**Summary Of The Review:**

I hold a double feeling of this paper. On the one hand, MixPro demonstrates its strong performance on various tasks and datasets., meanwhile, I like ideas that are simple yet sweet. On the other hand, I cannot agree with the explanation stated in the introduction and related work. I would first leave a borderline here and see how the authors respond.

---

> ### Author Response · Authors · 2022-11-19
> **Part 1 of Response to Reviewer vD1g**
>
>
> Thank you for carefully reading our submission and providing many insightful comments. We address the comments below:
>
> **Q1: However, the authors did not include the pseudo-code in their manuscript. It would be clearer if they can include the pseudo-code in the appendix.**
>
> **A1:** Thank you. We have added the pseudo-code in the appendix. We will also release the source code.
>
> **Q2: For region-based mixing, I am a bit confused about why region-based mixed images may provide insufficient image contents and why the proposed MaskMix can help to alleviate this problem as MaskMix will also mix just a portion of the entire image, which may be insufficient as well.**
>
> **A2:** The image mask of region-based methods is composed of only one continuous area for mixing original images, which may miss some important content of the object. In contrast, the image mask of our proposed MaskMix consists of multiple scattered and continuous areas in the image, which thus can capture broader parts of the object content of the original images from a global perspective.
>
> **Q3: For boundary artifact, the authors argue that it will introduce sharp rectangular borders that are clearly distinguishable from the background. However, the solution MaskMix will also lead to these sharp artifacts and the patch-wise mixture will spread this kind of noise everywhere in the mixed image, which may even aggravate such problem. Instead of echoing the argument by the authors, I would suggest that MaskMix may introduce more noisy patterns into the image mixture so that the network can be better regularized by these noises.**
>
> **A3:** TransMix introduces sharp rectangular borders even within an image patch (see Fig. 1, red colored) that are clearly distinguishable from the background. These borders inside the patch may draw excessive attention to the model, because for vision transformers, a patch would form a semantic unit, and the computation of the attention map of the whole image is obtained by summing the attention of each image patch.
>
> Our MaskMix is able to eliminate the influence of the internal borders of patches. By aligning the borders of the mask patches with the image patches, there thus will not be sharp rectangular borders within image patches. Thanks for your suggestion, we here also design three approaches to explore the performance of our MaskMix: A. Aligning mask borders to patch borders for TransMix. B. Misaligning the borders of mask patches with image patches for our MaskMix. C. Reducing the scale of each mask unit for our MaskMix to introduce more noises.  Experimental results are below:
>
> Trained with Deit-T on ImageNet-1k.
> |method| TransMix | A | B| C| MaskMix (ours)
> |-|--:|--:|--:|--:|--:|
> |Top-1 Acc| 72.6 | 73.1|  73.1|73.4 | 73.8|
>
> We can observe that our MaskMix introduces more noisy patterns (such as B and C) lead to worse results. The TransMix eliminates the internal borders of the patches using the approach A and lifting Top-1 Acc by \%5. This shows the improvements brought by the design in our MaskMix.

---

> ### Author Response · Authors · 2022-11-19
> **Part 2 of Response to Reviewer vD1g**
>
>
> **Q4: For Progressive Attention Labeling (PAL), the authors argue that it is better than TransMix since attention maps may not always be reliable during the training process. However, the $\lambda$ of TransMix is actually the mean of attention weights and the original $\lambda$ introduced by CutMix (a.k.a. $\lambda_{area}$ in the manuscript). This indicates that TransMix also considers this problem and uses the area of the cropped image as an extra condition. Meanwhile, TransMix also evaluates the effect of using different attention maps in calculating $\lambda_{attn}$. However, using a highly-discriminative attention map pre-trained by DINO to calculate $\lambda_{attn}$ does not result in any improvement in terms of the top-1 accuracy on ImageNet (see in Table 6 in TransMix paper). So I would doubt whether the label noise introduced in the label space would also be helpful.**
>
> **A4:** Thanks for the comment. To address this confusion, we need to clarify several issues: (Q4.1) DINO produces a highly-discriminative attention map on original images. However, vision transformer receives stronger data augmented mixed (such as Random Erase, Rand Augmentation and CutMix) images during training, can DINO produces a highly-quality attention map for these mixed images? How are the qualities of attention maps generated by different pre-trained models (such as DINO, TransMix, or PAL) for these mixed images? (Q4.2) What are the advantages and disadvantages of TransMix utilizing the mean of $\lambda_{attn}$ and $\lambda_{area}$. (Q4.3) What are the advantages of PAL compared with TransMix?
>
>
> > **A4.1:**  The pre-trained DINO produces a highly-discriminative attention map on original images, but unknown on stronger augmented mixed images. Thus, in this experiment, we aim to compare the quality of attention maps of stronger augmented mixed (such as Random Erase, Rand Augmentation and CutMix) images. The models compared to DINO are pre-trained vanilla DeiT, pre-trained TransMix-DeiT at 100/200/300epochs, and pre-trained PAL-DeiT at 100/200/300epochs. In detail, we first fine-tune the above-mentioned pre-trained models by conducting the Weakly Supervised Automatic Segmentation task on Pascal VOC 2012 benchmark for 20 epochs. Then, we calculate the Jaccard similarity between the ground-truth and segmentation masks obtained by thresholding the class-token attention $\textbf{A}$ from these fine-tuned models. The results of Jaccard similarity can imply the quality of attention maps (refer to Sec 4.4 of TransMix). For the training phrase, the model is weakly supervised since only the class-level ImageNet labels are used (i.e.  without per-pixel supervision for segmentation). For the validation set, we utilize stronger augmentation (Random Erase, Rand Augmentation and CutMix) to mix images and utilize CutMix to mix labels. Experimental results are below:
> >|method| DINO | vanilla-DeiT-T |TransMix-100epochs|TransMix-200epochs|TransMix-300epochs|PAL-100epochs|PAL-200epochs|PAL-300epochs|
> |-|--:|--:|--:|--:|--:|--:|--:|--:|
> |Jaccard similarity(\%)| 22.0 | 21.2| 14.7 | 22.3| 22.8 | 15.4|23.3 | 23.9|
>
> >We observe that the quality of the mixed images' attention maps generated from TransMix and PAL at 200 epochs are over DINO's. Furthermore, the quality of attention maps from PAL are better than the others after 200 epochs. This could also explain why TransMix slightly outperforms the result of utilizing DINO and the success of our PAL.
>
>
> > **A4.2:** TransMix considers unreliable attention maps and uses the mean of area of the cropped image as an extra condition. But this is not accurate and flexible enough. The advantageous aspect is that if TransMix receives a low-confidence attention map used for reassigning the mixed label, it introduces a mean of $\lambda_{area}$ for $\lambda$ which may provide compensation. The disadvantageous aspect is that if TransMix receives a high-confidence attention map, using a mean of $\lambda_{area}$ may lead to a bit of inaccuracy when adjusting the mixed-label assignment.
>
>
> > **A4.3:** Our PAL is more flexible and accurate for re-assigning the mixed label than TransMix, because it uses a dynamic progressive factor ($\alpha$), instead of a mean of  $\lambda_{attn}$ and $\lambda_{area}$.  If our PAL receives a high-confidence attention map, it would be more dependent on the $\lambda_{attn}$ according to Ep. (9) to re-assign the mixed label; otherwise, it would be more dependent on the $\lambda_{area}$.
>
>
>
> **Q5: The authors did a good job in describing their overall pipeline and the basic idea. However, I am a bit confused about whether the introduced image will be first shuffled or not before mixing. The cat in Figure 2 does not follow the original geometric structure, but there seems to be no shuffling operation in Eq. (7).**
>
> **A5:** We do not shuffle the geometric structure of original images. We will make this more clearly in paper.

---

> ### Author Response · Authors · 2022-11-19
> **Part 3 of Response to Reviewer vD1g**
>
>
> **Q6: My assumption is that MixPro creates more complex scenarios in the image mixture such that the network can be better regularized. Thereby I am curious about whether introducing more random patterns will further improve the performance. These random patterns may include but not be limited to: (1) random shuffling (or shifting) on the cropped patches. (2) random shuffling (or shifting) on the patches of the canvas image**
>
> **A6:** Thanks for the suggestion. We test random patterns to evaluate the DeiT-T on ImageNet, and the results are below. We can observe that damaging the original geometry of the image with these schemes leads to worse results, i.e., simply introducing random patterns (e.g., via simple random shuffling) does not bring performance improvement.
> ***
> |model |results|
> |-|--:|
> |baseline| 73.2 |
> |random shuffling on the cropped patches| 69.8 |
> |random shuffling on the patches of the canvas image| 69.7|
> |Both| 67.3 |
> ***
>
> **Q7: The proposed method is only applicable to Vision Transformers with class token.**
>
> **A7:** Thanks for the suggestion. We also show the results on token free Vision Transformers (such as Swin) in Tab. 1. This experiment follows the work of TransMix employing classification attention (CA) block for Swin without parameter overhead to get attention map. Another approach is to utilize tokens-to-tokens attention approach [1] to identify the attentive tokens, where the attention scores are computed as the average of the attention from all tokens to all tokens. These results are below. This indicates that the method can also be used for Vision Transforms without class token.
> ***
> |model |baseline|+TransMix|+MixPro|
> |-|--:|--:|--:|
> |CA-swin-T | 81.6 |  81.8 |82.8 |
> |tokens-to-tokens| 81.6 |  - |82.6|
> ***
>
> **Reference**
> [1] Liang Y, Ge C, Tong Z, et al. Not all patches are what you need: Expediting vision transformers via token reorganizations[J]. arXiv preprint arXiv:2202.07800, 2022.

---

> ### Comment · Reviewer_vD1g · 2022-12-12
> **Final decision**
>
> I appreciate the author's efforts in clarifying everything and the author's response basically resolves my concerns except for 1.1 and 1.2 and 4. I lean to accept the paper due to its efficacy and simplicity. However, the authors should update Figure 1 accordingly in the manuscript and include results reported in A3 and A6.

---

### Official Review · Reviewer_qEAs · 2022-10-25

**Confidence:** 4
**Correctness:** 4
**Technical Novelty And Significance:** 3
**Empirical Novelty And Significance:** 4
**Recommendation:** 8

**Clarity, Quality, Novelty And Reproducibility:**

The paper is well written and easy to follow. The authors have conducted plenty of experiments to demonstrate the effectiveness of the proposed method and the proposed approach can be reproduced based on the information given in the paper.

**Strength And Weaknesses:**

Pros:
1. The paper is well written and easy to follow.
2. The experiments are thorough and the results are pretty good.
3. The proposed method can be generalized to different visual tasks and boost performance.
Cons:
1. In the downstream tasks, seems like the proposed method is only employed for pretraining. Can it be used for directly finetuning for downstream applications?
2. Minor: in Generalization analysis: "Fig2(c) depicts their loss and top-1....." should be Fig2(c) and (d) depicts their loss and top-1...."

**Summary Of The Paper:**

In this submission, the author proposed a data augmentation method for ViT. Specifically, two samples will be mixed in a global level, where patches of an image will be replaced with the patches from another images at corresponding locations. Then, the labels are generated using both the attention map generated by the ViT and the binary mask to avoid the negative effect using only attention map in the beginning of training process. The experimental results have illustrated the effectiveness of the proposed method. The approach is further demonstrated in the downstream applications.

**Summary Of The Review:**

To sum up, the paper is well written and easy to follow. It proposed a data augmentation method for ViT which can further boost the performance of ViT on various tasks. Experiments on different tasks and applications were conducted and the results were well analyzed. I have some questions which can be referred to in Strength And Weaknesses.

---

> ### Author Response · Authors · 2022-11-19
> **Response to Reviewer qEAs**
>
>
> Thank you for the positive and helpful comments. We address the concerns below:
>
> **Q1: In the downstream tasks, it seems like the proposed method is only employed for pretraining. Can it be used for directly fine-tuning for downstream applications?**
>
> **A1:**  Yes. We fine-tune DeiT-S for long-tailed recognition on Places-LT [1]. In detail, we fine-tune DeiT-S for 20 epochs via logit-adjusted loss [2]. Experimental results are below:
>
> |method| DeiT-S| MixPro-DeiT-S
> |-|--:|--:|
> |Top-1 Acc| 35.3|37.1|
>
> **Q2: "Fig2(c) depicts their loss and top-1....." should be Fig2(c) and (d) depicts their loss and top-1...."**
>
> **A2:** Thanks for pointing this out. We have corrected the typo in the revision.
>
> **Reference**
>
> [1] Ziwei Liu et al. Large-scale long-tailed recognition in an open world. In Proceedings of the IEEE/CVF Conference on Computer Vision and Pattern Recognition, pages 2537–2546, 2019.
> [2] Aditya Krishna Menon et al. Long-tail learning via logit adjustment. arXiv preprint arXiv:2007.07314, 2020.

---

### Decision · Program_Chairs · 2023-01-20

**Decision:**

Accept: poster

**Justification For Why Not Higher Score:**

The novelty of the proposed approach is somewhat limited.

**Justification For Why Not Lower Score:**

The proposed methods extends/specifies existing approaches to vision transformers and the achieved results are promising. The paper is well written.

**Metareview: Summary, Strengths And Weaknesses:**

This paper proposes a data augmentation method for ViT, where two samples are mixed on a global level, by mixing patches between the images This is strongly related to previous data augmentation methods like cut-mix for  convolutional neural networks. The experimental results show that the proposed method is effective.

Overall, the proposed ideas are simple and interesting, the paper is well written. Yet, the novelty is limited and the results could have been validated on more datasets. Overall, the evaluation is however, still sufficient.

**Note From Pc:**

if the above contains the word "oral" or "spotlight" please see: "oral" presentation means -> notable-top-5% and "spotlight" means -> notable-top-25%. As stated in our emails, we are disassociating presentation type from AC recommendations